# Developing a Colorimetrically Balanced, Measurement-Based Petal Colour System for Cultivated Rose (*Rosa* L. Cultivars) and the Resulting Colour Categories

**DOI:** 10.3390/plants13101368

**Published:** 2024-05-15

**Authors:** Gábor Boronkay, Dóra Hamar-Farkas, Szilvia Kisvarga, Zsuzsanna Békefi, András Neményi, László Orlóci

**Affiliations:** 1Institute of Landscape Architecture, Urban Planning and Garden Art, Hungarian University of Agriculture and Life Sciences (MATE), Páter Károly út 1., H-2100 Gödöllő, Hungary; hamarne.farkas.dora@uni-mate.hu (D.H.-F.); kisvarga.szilvia@uni-mate.hu (S.K.); nemenyi.andras.bela@uni-mate.hu (A.N.); orloci.laszlo@uni-mate.hu (L.O.); 2Institute of Horticultural Sciences, Hungarian University of Agriculture and Life Sciences (MATE), Páter Károly út 1., H-2100 Gödöllő, Hungary; kovacsne.bekefi.zsuzsanna@uni-mate.hu

**Keywords:** cultivated rose, petal colour, CIEDE2000, chromatic difference, colour distance, classification

## Abstract

There is no practical and at the same time objective colour system available for describing cultivated roses (*Rosa* L. cultivars). For this reason, a new colour classification system was developed which is colorimetrically balanced and appropriate for algorithmic colour identification; however, it is also suitable for field-work. The system is based on the following colorimetric criteria: (A) Each colour category is characterised by a measured petal colour in the CIE L*a*b* standard as the centroid of the category. (B) The CIEDE2000 colour differences between the adjacent centroid colours are limited (5 < ΔE_00_ < 7). (C) The maximal colour difference between the measured colours in a category is also limited (to 12.12 ΔE_00_). (D) A measured petal colour can only be classified into an existing category if the colour difference from the centroid colour of the given category is less than 5.81 ΔE_00_, otherwise a new category is required. (E) A category is only considered non-redundant if it has at least one measured petal colour that cannot be classified elsewhere. (F) The classification of the petal colours is based on the least colour difference from the centroid colours. As a result, 133 colour categories were required for describing all the 8139 petal colours of the rose cultivars of the Budatétény Rose Garden (Hungary). Each colour category has the following parameters: standardised colour name, the colorimetric parameters of the centroid, grouping, RHS colour chart coding, and reference cultivars, which are described in the article.

## 1. Introduction

Reliable cultivar identification is fundamental in ornamental plant collections, especially in the case of mass propagation. From this point of view, one of the most problematic horticultural groups is the cultivated or garden rose (*Rosa* L. cultivars), since the phenological variability of this plant is exceptionally high. This is due to the fact that the cultivars are complex combinations of alloploid species [1] and the varieties are the results of more than five thousand years of breeding [2]. In some cases, the varieties are hardly distinguishable from each other, and sometimes the difference is so huge, as if they were two species, which makes the comparisons difficult.

Variety identification is impossible outdoors (in situ) without any appropriate variety description. In the case of roses, this is especially true for the flower colour. However, the breeder’s descriptions of the flower colouration in the variety registrations [3] do not prove to be suitable for identification work, because the elaboration of the cultivar descriptions is extremely different. Unfortunately, the colour standard of the ARS (American Rose Society) and WFRS (World Federation of Rose Societies) uses only 18 categories [4] which is not enough to distinguish the varieties. The UPOV (Union Internationale pour la Protection des Obtentions Végétales) standard [5] proved hardly better for in situ field-work. The description method of the petal colouration of the UPOV standard is based on the printed colour card collection of the Royal Horticultural Society (RHS colour chart) [6]. Unfortunately, this set of colour tags arranged in fans is only partially suitable for colour identification in the case of roses. The usability of the colour system of RHS is limited by the fact that this colour set is not balanced colorimetrically (the colour differences between the adjacent printed colours are very different), the colours are not defined colorimetrically (there are no printed colorimetric values), the chart lacks several highly saturated colours, and the colours are identified by number-letter codes instead of natural colour names or colorimetrical parameters. Since petal colours and card colours are almost never identical, the correct description of petal colour requires interpolation (if a petal colour seems to be mixable from card colours) or extrapolation (if a petal colour is more or less saturated, lighter or darker than the RHS card with the closest colour to it). These mathematical functions need exact colorimetric parameters, but the factory colorimetric data of the RHS colours are missing.

The literature on the subject is rather poor, since, as far as the authors know, no one has yet created any colour system for ornamental plants based on scientific principles. In rose cultivation, there are only two standards for breeders and traders, and both were and are used in the cultivar registration by the American Rose Society (ARS). The first one, the Horticultural colour chart I–II [7] is one of the oldest horticultural colour standards ever printed, which was published 80 years ago, and it can be considered the top quality of the printing technology of that time. This book is still popular and used—although only informally—because, in addition to the standardised colour names of the textile industry and trade, the pages also include detailed explanations. The biggest problem of this book is the moderate number of colours, and the aged and discoloured printing inks. The second standard is the official, 18-category colour system [4] of ARS; however, less than 20 colours is hardly enough for an authentic cultivar description.

The number of sources is also very few in the case of practical colour standards. Despite the fact that there are hundreds of industrial colour systems and colour standards, only a few of them are both comprehensive and have named colours. Probably the most widely used of them are the Pantone [8], Crayola [9] and RAL [10] standards; however, the colours of the printed Pantone Formula Guide are unnamed [11]. It does not make the job any easier that the exact colorimetric parameters of the colours of these industrial standards are not public and the colour names of these systems are usually fancy ones, not the names of well-known coloured natural objects. One of the exceptions described by Séve [12] is the book by the same author [13], where each colour hue is classified into subcategories by saturation and lightness.

Although several publications can be found on plant colours, these works primarily focus on the chemical and plant physiology background of the colour pigments, and do not provide assistance for practical colour identification. Brubaker [14] gave a good overview of the colours found in roses especially for non-professionals, although, for this reason, his paper lacks a mathematical background. De Vries et al. [15] showed the inheritance and breeding role of carotenoids, anthocyanins, and flavonoids, where the authors focus on pelargonidin. In their later publication [16], the genetic background of these pigments was investigated. The biochemistry and genetics of the development of pigments were studied by Ogata et al. [17], while Uddin et al. [18] examined the role of light and sugar in the gene expression of flower pigments, although their subject was lysianthus (*Eustoma russellianum*) not rose. Gonnet [19] evaluated the relationship between the anthocyanin content of petals, co-pigments, and CIE colour standards, while Mol et al. [20] examined the plant pigments in general. Ferrante et al. [21] studied the pattern of flower colour and petal senescence, while Eugster and Fischer [22] published the chemical background of the pigments. More attention has been paid to the pigments containing delphinidin required for transgenic blue roses and to the genetics of these pigments, and these include the publications of Gonnet [23], Katsumoto et al. [24], Fukui et al. [25], Urs et al. [26], Sasaki and Nakayama [27], and Dalrymple et al. [28].

It seems surprising that there is a relatively small number of studies on the social role of rose flower colour because it seems to be a very important trait. The colour of the flower is the main consideration when buying a rose, although roses were originally cultivated for their fragrance. Shopping habits [29] were investigated in Germany, as it is the largest host market in the European Union. Here, the most common colours of solid bouquets were red and yellow (24% and 23%), but yellows and orange–salmon colours gradually became more popular compared to the shades of red and pink.

The psychological value of petal colours is unique to the individual, but is strongly influenced by current fashion trends. Cultivation tries to satisfy this immediate need, so according to surveys, in addition to psychological factors, marketing and financial aspects also play a role. There was a survey in South Africa [30] that specifically looked at whether people buy different colours or types of flowers for different emotional reasons (gifts, hospital visits, and homes). However, no clear pattern emerged from the results. Neither the type of flower nor the colour was of primary importance when purchasing. Apart from personal taste, the customers only took into account that the colour of the flower should be clean and bright. However, this was probably driven by fashion, as later soft colours (romantic) and gradient petals became popular.

In contrast to the small number of independent publications, many works by the authors of this paper show the development process of the petal colour classification system presented here. The dynamics of the fading process during the life of the rose flower was published in 2009 by Boronkay et al. [31], and later the evaluation of measured colour differences between different flower phenophases and rose cultivars was published [32]. The rose petal colour system and its special problems were published by Boronkay first in 2016 [33], and then it was presented graphically [34].

Since it was not possible to find an appropriate colour system for rose petals for field-work, it seemed necessary to develop a colorimetrically balanced classification system. The following expectations can be formulated for an ideal petal colour system: It should be objective: the colour categories must be based on instrumentally measured data and should be characterised by reference cultivars. It should be colorimetrically balanced: the number of colour categories, and their location in the colour space (3D colour model) must be specified by objective colorimetric rules. It should be practice-oriented: the resolution of the system (numbers of the colour categories) cannot be so high that they cannot be recognised in field conditions. The colour categories must have memorable names, and the denomination system must be standardised also.

Based on the articles listed above, the colorimetry-based rose petal colour system created by the authors can be considered a completely independent development and is not related to any project known to the authors. Although the colour categories of the rose-petal colour description system can be considered the final result of the project, the novelty value lies primarily in the method by which colorimetrically balanced petal colour system can be created from a mass of unstructured colorimetric data. That is why the steps of creating this colour system will also be presented as a part of the result, not as a method.

The expectations towards the petal colour system can be described with the following rules:The colour system should be a set of colour categories characterised by a typical reference colour: the so-called centroid colour.The centroid colours should be measured as the petal colours of the reference varieties.The classification method of petal colours is the least colour difference from the centroid colours.The colour differences between the adjacent centroid colours should be limited.The centroid colours should evenly fill the part of the colour space in which rose petal colours can occur.The number of categories should be optimised by colorimetric calculations.For everyday field-work, the petal colour system should have an easy-to-remember and regulated nomenclature for the categories.This system should be suitable for algorithm-based automated classification.

Based on the rules, the categories of such an ideal colour system are located in a colour space in a balanced manner. However, petals cannot take any colour due to the physical structure and biochemical conditions of the organ, so the placement of centroid colours in the colour space cannot be perfectly regular. Because of this limitation, the colorimetric balance of the centroid colours, and consequently, the shape of the categories surrounding the centroids cannot be perfect. When designing the system, it must also be taken into account that the reference cultivars should be well known as much as possible, and the petal colours of these reference cultivars should be well recognisable. Because of these factors, creating such a colour system requires a multi-step process and a weighted consideration of factors.

## 2. Results

### 2.1. Defining the Basic Colour Categories and Developing the Framework of the System

Used rules: rule 1: the colour system is a set of colour categories characterised by a typical centroid colour; rule 2: the centroid colours are measured petal colours of reference varieties; rule 3: the classification of the petal colours is based on colorimetric calculations.

The schematic diagram of the development of the colour system of the rose petals is shown in Figure 1. In 2004 as a pilot, all the flower colours in the Budatétény Rose Garden (Budapest, Hungary) and Gergely Márk’s breeding garden in Törökbálint (Hungary) were evaluated by visual comparison. Altogether, 1650 cultivars were assessed, and they were classified into typical colour categories based on subjective, personal opinion. Then, in each colour category, a reference cultivar was chosen, whose colour seemed appropriate to characterise the average colour of that category. The petal colours of the reference cultivars were measured with a spectrocolorimeter in the CIE L*a*b* colour space, and these reference colours were considered the initial centroid colours of the categories (each category has only one centroid colour).

At the actual start of the project, all the characteristic colours of the items of the Budatétény Rose Garden were measured in situ by a spectrocolorimeter (10 samples/colour, see Section 4.4, Section 4.5 and Section 4.6 “Materials and Methods”). Even though the number of the colour categories and their spatial location in the colour space were still based on personal considerations, in this phase the petal colours were already classified into categories by colorimetric calculations. The classification was based on the least colour difference between the given colour and the centroid colours (measured in ΔE_00_ of the CIEDE2000 (or CIE ΔE_2000_) colour difference standard). As colour difference cannot be a negative number, the least sum of squares calculation was unnecessary.

This classification needed the huge colour difference matrices of CIEDE2000 colour differences between all measured colours and all centroid colours. As the number of categories and the colorimetric parameters of centroid colours were changed in further refinement, these classifications had to be performed several times. As the maximum number of the colour categories was 133, and 8139 measured colours were classified, this procedure needed extremely large CIEDE2000 matrices with 1,082,487 cells. At that stage of data processing, all the measured petal colours were classified into categories based on colorimetric calculations; however, the positions of the centroid colours in the colour space were not balanced yet.

### 2.2. The Balance of Centroids and Colour Categories in the Colour Space

Used rule 4: the colour differences between the adjacent centroid colours are limited.

The balanced distribution of colour categories in the part of space where petal colours can occur is based on the fact that the colour differences between the adjacent centroid colours are limited. Although the degree of this colour difference fundamentally affects the construction of the colour system, there were no preliminary data on how much colour difference is ideal. However, it was expected that the colour system should have a good resolution, but at the same time, the adjacent colours could be separated visually even in situ.

Having searched for the optimal colour difference, CIEDE2000 values were calculated between easily distinguishable characteristic petal colours that were selected visually (Table 1). The average CIEDE2000 colour difference between these adjacent, but visually easily separated colour categories, was found to be near 6 ΔE_00_, although the variance of values was extremely high, about 3.1 < ΔE_00_ < 8.6. Based on this comparison, the appropriate distance between the centroid colours is 6 ΔE_00_, which ensures that the resolution of the system is optimal: the number of categories is as high as possible, while they can still be separated in the field. However, this rule cannot be strictly followed, because the variability of rose petal colours is not infinite, and sometimes no measured example meeting this condition can be found. To find the strictest, but still usable limitation, the maximum and minimum of the allowed colour difference between the centroid colours were tightened step by step, and new centroid colours were calculated which fulfilled the rule as long as appropriate petal colours were found. As a final result, the 5 < ΔE_00_ < 7 colour difference range seemed the strictest but sufficiently flexible limitation. Although 5.5 < ΔE_00_ < 6.5 were the target values, having processed 80,000 measured petal colours, it was realised that the arrangement of the rose petal colours in the CIE colour system was not so regular.

To develop a balanced set of centroid colours based on the above-mentioned limitation, colour difference matrices were created between all the centroid colours. In the final stage, the CIEDE2000 colour matrix between 133 colours meant 17,566 (133 × 132) ΔE_00_ values. For each category, the minimal ΔE_00_ colour difference was looked for, and all the centroid colours were replaced where the minimal colour difference was too small or too high (ΔE_00_ < 5 or ΔE_00_ > 7). If the colour difference was too low, the problem could be solved by excluding a colour category or by merging colour categories. If the colour difference was too high, new colour categories had to be chosen from the measured petal colours according to the set of rules described in Section 4.7 “Materials and Methods”.

In this step of the project, a net of colours was formed where the colour difference of the adjacent centroid colours was limited, although this is not yet a guarantee of perfect balance. The missing and redundant colour categories can still distort the balance of the petal colour system.

### 2.3. Finding the Discontinuities between Colour Categories

Used rule 5: the centroid colours evenly fill the colour space.

A limited colour difference between the centroid colours (Rule 4) is not a guarantee of complete balance because it is not enough to eliminate the phenomenon when two or a small number of colours form separated groups.

To eliminate the discontinuity when a colour twin forms an independent group far from the other colours, not only the smallest colour difference between each category must be limited, but the second smallest one should be limited also. According to this rule, each colour category should be close (5 < ΔE_00_ < 7) to at least two of its neighbours, not only to the closest one. However, the colour space formed by the petal colours is not a perfect body of rotation, and there are some corners and edges where the most extreme colours are situated (e.g., #1 petal white, #49 chrome orange, and #97 manganese black). In the case of these colours, only one neighbouring colour is possible, so the rule should be redefined as each colour category should have at least one neighbouring colour which is close to at least two adjacent colours within the 5 < ΔE_00_ < 7 limit. This modified rule excludes the isolated subgroups of twin colours in the net of centroid colours. However, this rule does not solve the problem when more than two colours form a separate group; see Figure 2 as a non-real, but easy-to-interpret example.

For spotting any gap between the centroids, an animated 3-dimensional point matrix was drawn from the centroid colours in the CIE L*a*b* space by a statistical software. As the point matrix is rotated around the axes, the empty spaces between the points can be searched visually. Since the colour categories cannot fill the whole CIE L*a*b* colour space, only the part of the colour space drawn by the possible petal colours was checked. Within this part of the space, the centroid colours should be distributed more or less homogeneously. According to the 3-dimensional point matrix, the colours of the rose petals—at least the centroid colours—form a candy cane or a 3D check mark shape in a CIE L*a*b* colour system (Figure 3).

By rotating the point matrix, the empty gaps where a centroid is possibly missing are noticeable. Although this method proved useful, it lacks objectivity and a 3D colour space on a 2D chart can also sometimes be misleading.

### 2.4. Colorimetric Detection of Missing Colour Category

An additional and much more exact solution was needed. If the colour categories fill the colour space more or less in a balanced way, the sizes of the colour categories are more or less the same. By measuring the size of the colour categories, the inconsistency of the centroid colour net can be revealed. Two questions arise here: (1) how can the size of a colour category be measured, and (2) when is a colour category considered too large?

The size of a colour category can be estimated if the location of the measured colours within the category is known, and the number of those colours is not too small. The biggest colour difference between the measured colours within a category can be considered the longest dimension of the category. If the biggest difference is too high, it indicates an unreasonably large colour category in the colour space; therefore, a colour category is probably missing here, and this too-large category should be separated into two or more new categories.

The problem of the maximum allowable colour difference within a category is more difficult. The shape of the category is determined by the set of points that are closer to the category’s own centroid colour than to the other centroid colours. Unfortunately, the CIEDE2000 colour difference standard is a non-Euclidean function; therefore, the theoretical border between the adjacent categories is irregular and practically unpredictable.

To eliminate this difficulty, a simplifying model was created, where the locations of the centroid colours in the CIE L*a*b* colour space are perfectly regular, as the colour differences between them are constant. Such an arrangement can be imagined as a cubic crystal system, where the colours represent the vertices of regular hexahedrons. The lengths of the edges of the cubes are uniformly 7 units, as this is the maximum allowable colour difference between two adjacent centroid colours (according to Section 2.2). However, this model also needs a linear colour difference standard. Such a CIE standard exists, although it is considered to be outdated; it is the CIE76 [35] standard, where the dimension of the colour difference is ΔE_76_. In this model, the centroid colours form a 7 ΔE_76_ side length regular hexahedron in the CIE L*a*b* colour space. In this regular arrangement, the shape of the colour category (the part of colour space closest to the centroid) can already be interpreted, which also forms a 7 ΔE_76_ side length regular hexahedron. In this simplified model, the largest possible colour difference between two colours in a colour category (the longest dimension) can now be calculated. The maximal distance between the furthest points of this polyhedron is the body diagonal of the hexahedron: D = (3 × a^2^)^1/2^, in this model D = (3 × 7^2^)^1/2^ = 12.12 ΔE_76_. According to this simplification, the maximum acceptable dimension of a colour category is 12.12 ΔE_76_. Assuming that the colour distances calculated according to the CIE76 (ΔE_76_) and CIEDE2000 (ΔE_00_) standards are not too different (the main difference is in their linearity), 12.12 ΔE_00_ (instead of ΔE_76_) can be considered the maximal allowable colour difference in a petal colour category. Accordingly, when a larger than 12.12 ΔE_00_ colour difference was found within a category, this category would be split into two or more smaller ones. When creating these new colour categories, the rules written in Section 4.7 “Materials and Methods” should be used.

### 2.5. Optimising the Number of Colour Categories

Used rule 6: the number of the colour categories is optimised by colorimetric calculations.

A petal colour standard cannot be a closed system, because there is always a chance that a novelty cultivar appears with a completely new petal colour that cannot be classified into any existing category. In such cases, a new category should be introduced for this colour where the new centroid colour will be the petal colour of the novelty. The question is how big the colour difference between a measured colour and its nearest centroid colour should be to be considered a new one. As the optimal colour difference between the neighbouring centroid colours is 6 ΔE_00_, this difference seems to be the appropriate value. If the colour difference between the new colour and any existing centroid colour is smaller than 6 ΔE_00_, the new colour should be classified into this colour category, while if even the smallest colour difference is higher than that, a new colour category should be formed for the new colour. If this least colour difference is 7 ΔE_00_ or higher, two or more new colour categories should be created, as the maximum allowed colour difference between centroid colours is 7 ΔE_00_.

Although this method seems to control only the expansion of the classification system, it is also appropriate for optimising the number of categories, ensuring that there are no unnecessary (redundant) colour categories in the system. Here, “unnecessary” means that even if a category were eliminated, all the colours belonging to it can be classified into another category, since they are also close enough to other centroid colours as well (ΔE_00_ < 6). If this is true, this category should be excluded as a redundant one. Colour categories are only considered necessary if at least one measured colour can be found which is close (ΔE_00_ ≤ 6) exclusively to this colour category and only this category. However, as detailed in Section 2.6, this 6 ΔE_00_ value should be reduced a little, because the inaccuracy of the sampling can affect the number of the categories to be excluded.

### 2.6. Solving the Inaccuracy of Measurement

It has been experienced that by taking rule 6 strictly (see Section 2.5), even some potentially significant and characteristic colour categories had to be excluded from the colour system as redundant ones. That was the case with the category of #97 manganese black, #103 Neyron rose, #102 beetroot purple, etc. Examining the situation, it was found that the colours originally classified in these categories could be classified in other categories, but almost only by accident. The colour difference between these colours and their second closest centroid colour was just a little lower than 6 ΔE_00_. It is possible that with a small change to the data recording—for example, a microscale deviation of the measurement point—these colour categories would still be needed.

It means the parameters of the measured colours might be distorted by some measurement inaccuracy that results from the methods of sampling. From a practical point of view, the effect of an additional redundant colour category is less harmful than an unreasonably omitted one. To solve this problem, it seemed worthwhile to reduce the colour difference limit (ΔE_00_ = 6) of Section 2.5 by a very small tolerance value. If the limit is smaller (the rule is stricter), some measured colours will no longer be close enough to the second-closest colour category.

As a result, some of the categories that seemed to be unnecessary should not be omitted, so they can still be a part of the system. Due to the fact that the CIEDE2000 colour difference is a non-linear method, the uncertainty of sampling can only be estimated. In the absence of an official solution, the data variability was computed within the measured colours (10 samples) and 10% of this variability seemed to be appropriate for determining this estimated inaccuracy. The variability calculation was based on the sample standard deviation formula, but instead of the “x_i_ − $\bar{x}$” part, the “ΔE_00_” colour difference was used. The new, unofficial formula is as follows: S_ΔE00_ = (Ʃ (ΔE_00i_^2^)/(N − 1))^1/2^. According to the calculation, 10% of the median of the S_ΔE00_ value is 0.19 ΔE_00_ as the quantity of “uncertainty”. Using the median instead of the average was advisable because the average of the standard skewness was very high: Sk = 2.87. Accordingly, the limit of entering a colour category into the system should be reduced, and 5.81 instead of 6 ΔE_00_ colour difference is the optimal limit. This limitation factor should be used for both inserting a new category and selecting unnecessary categories. According to recalculation, this small change significantly reduces the number of categories considered redundant.

### 2.7. Grouping the Colour Categories

In order to organise the colour system, the colour categories were combined into colour groups using Ward’s method of cluster analysis. As no optimal group number could be found with mathematical statistics (Agglomeration Distance), 28 groups were created for practical considerations. The clustering process was based on five variables: CIE L*, a*, b*, C*, and h_33_* (see Section 4.9 and Section 4.10 “Materials and Methods”) parameters of the centroid colours. All the 28 groups were given a name, although without the strict naming rules written in Section 4.12 “Materials and Methods”. The nomenclature of the groups can be seen in Table A1, Column 1).

### 2.8. Colour Names

Used rule76: for everyday field-work, a regulated nomenclature is required for the categories.

According to the hypothesis, the naming of the centroid colours has fundamental importance. However, developing a colour standard needs significant professional consensus, so the names of the colour categories created by the authors must be considered “suggested names” only.

The ideal colour naming process would be a software-aided calculation based on minimal colour differences between the category to be named and the colours of an industrial colour system, where both names and colorimetric parameters can be found. For this purpose, HTML colours [36], the Crayola standard colours [37], the Pantone TCX colours [38], the RAL Classic colours and RAL Design System+ [39] were used (Table A2). Unfortunately, none of these industrial colour standards gave satisfactory results, as the spatial distribution of their colours and the terminology of the colour names was extremely irregular. It became clear that a suitable name had to be found for each colour separately and manually.

A strict system of criteria was established for the colour names, which is described in Section 4.12 “Materials and Methods”. Having applied these rules, many well-known, well-defined colour names have to be omitted because they are just between two existing colour categories, such as cherry red. Elsewhere, due to the rules, it was necessary to find a new name, as the adjective part of the name was used already (cadmium yellow—cadmium orange—cadmium red), which is not allowed according to the rules created by the authors. In some cases, a colour name had to be omitted due to linguistic definition difficulties, for example, carmine red or lemon yellow are names of very different colours. In order to avoid the difficulties mentioned above, rare, lesser-known names had to be used in some cases. The best example is the #115 sea pink rose which does not mean “pink like the sea” but the “colour of the flower of sea pink (*Armeria maritima*)”.

The primary source of the names was the British Colour Council colour standard [7], although this means hardly 200 named colours. When no suitable colour name could be found, hundreds of Internet pages (encyclopaedias, fashion and beauty product web shops, and paint and dye catalogues) were browsed for finding a colour name which seemed both linguistically and visually appropriate. However, the majority of commercial names are fancy names, which were omitted, because they are practically impossible to interpret (e.g., pink panther or blossom beauty). The best online colour collection seemed to be Encycolouropedia [40] where 294 industrial colour systems can be found and compared.

The names suggested by the authors and the numbering system of the categories are presented in Table A1 with some name explanations. Even so, the colour names of this petal system cannot be perfectly objective, therefore each group was given a number also. However, the everyday field-work needs easy-to-remember description systems, which is why the petal colour categories were named, but this nomenclature is definitely just a recommendation, although a carefully considered one.

### 2.9. Using the Petal-Colour System by Software-Controlled Classification

Used rule 7: the petal colour system is suitable for algorithm-based automated classification.

Since colour identification needs huge CIEDE2000 colour difference matrices, special colour classification software for outdoor work seems very useful. For this purpose, a spreadsheet-based software was developed for MS Windows and Android operating systems. This software is able to classify any measured petal colour into one of the predetermined colour categories, or send a warning if the colour is unclassifiable as even the smallest CIEDE2000 colour difference is higher than 5.81 ΔE_00_ between the measured colour and the centroid colours. The software will be available only when the petal colour system has been published.

## 3. Discussion

With the help of the newly developed colour system of rose petals, all the 8139 recorded colours of the roses in Budatétény Rose Garden could be classified. This classification needed 133 colour categories, which are presented in Table A1, Table A3 and Table A4, and Figure 4. The number of categories was determined by the colour rules used and the spectrum of the actual measured petal colours, and subjective decision practically did not play a role in this. Since the Budatétény Rose Garden has a significant number of rose cultivars, several of them have special colours, and the colorimetric data were recorded even on the unopened flowers; this petal colour system covers a much wider colour range than the colorimetric variability of everyday commercial rose cultivars. For example, colorimetric data were recorded from the dark purple-violet (“Midnight Blue”), yellow ochre (“Honey Dijon”), purple-brown (“Hot Cocoa”), greyish lavender (“The Scotsman”), light brown (“Café”), dark yellow (“Sunsprite”), orange (“Magic Lantern”), and very dark red (“Taboo”) petals.

Considering that rose petals are moderately translucent (especially the white, light yellow, and blush ones), the measured colours are laden with a small amount of data bias due to the increasingly waxy white plastic labels used as petal support during the measuring process. According to some post-tests, the maximal effect of this wear was 33.9% of the plastic label discolouration on white petals and virtually no effect on the dark (red and purple) petals. Accordingly, the estimated maximal effect was about 0.45 ΔE_00_, which is practically negligible.

Comparing these 133 colour categories to the industry colour standards, the number of categories does not seem too high. For instance, the Sixth Revised Edition of the RHS colour chart (a standard specifically optimised for plant colours) provides 920 colours, although there are hard-to-distinguish colours and colour gaps can be found. However, the number of physically detectable colours actually occurring on a rose petal cannot be defined; it is practically infinite, and only the physiological processes of colour perception determine how many petal colours a person can distinguish. That is why it seemed necessary to set up objective colour categories, where the colours can be easily separated from each other, yet the system has sufficient resolution to separate the rose cultivars.

Although the data were recorded exclusively in the Budatétény Rose Garden, the database can be considered representative due to the diverse plant material of the rose garden and the extensive data collection. Based on this, the following statistical results acceptably reflect the general colour distribution of the petals of cultivated roses.

The lightest colour category in this colour system is #1 petal white (L* = 94.4), although it is far from absolute white. The darkest colour category is #96 raisin black (L* = 13.7), even though #97 manganese black seems visually darker, because the saturation of the latter colour is lower. The #24 cobalt yellow category shows the highest colour saturation (C* = 101.7), which proves the everyday experience that yellow and orange-red are usually the most vivid (saturated) petal colours in nature. However, the most conspicuous parameter of a colour is the hue. According to the database of the Budatétény Rose Garden, the coldest, most bluish hue (the highest negative h_33_* value) belongs to the #119 African violet category (h_33_* = −64.5°). The opposite extreme hue would be the highest h_33_* value, but this greenish-yellow category is actually the #1 petal white (h_33_* = 73.5°). For the explanation of h_33_* as a modified CIE h parameter see Section 4.9 “Materials and Methods”.

Significant differences could also be found in the distribution of colours. According to the data, the highest frequent colours were about 3% of all recorded petal colours (Table 2). These are the #83 Bengal red (3.2%) and the #1 petal white (3.0%). Most of the frequent colours are found on the abaxial surface of the petals, such as #103 Neyron rose and #85 schiller red (both 2.9%), because the colour variability of the back of the petals is poorer than that of the upperside.

A total of 23 most frequent colours cover 50% of all the measured colours, while the least common 48 colours cover only 5% of it. This unbalanced frequency suggests that any petal colour system should be open for new colours, because there is always a chance that an unusual-coloured cultivar appears. An ideal example of this phenomenon is the “Honey Dijon” rose cultivar, because its mustard yellow colour is completely unique and four categories are based on the colours of this variety (#11 sesame seed yellow, #15 ochre yellow, #26 tan yellow, and #27 Dijon yellow).

Finding the most appropriate names for the petal colour categories was a very time-consuming task. Using industrial colour systems proved unsuitable, although several standards had been tried (Table A2). Although these standards are provided with CIE colorimetric parameters and assigning a name to a centroid colour could be automated by the calculation of colour differences, the result was poor. Sometimes the minimal CIEDE2000 difference between the centroid colour and any colour from the standard was significantly higher than the ΔE_00_ = 5.81 limit, and sometimes the same colour was the closest one to several categories. Their nomenclature is based on fancy names, and these cannot be associated with any well-defined colour. That is why naming each colour category needed very careful and precise work.

Considering that no colour classification system is known that is based on the measured colours of a plant organ and regulated by colorimetric rules and conditions, in the absence of any template a totally new method had to be developed and the majority of the rules had to be created and tested during the creation process. Testing the finished petal colour classification system by algorithm-assisted classification and by everyday field-work has proved that this classification system is useful and meets the preliminary expectations. The set of rules established in advance also appears applicable: According to rules 1–3 a set of colour categories could be created with central (centroid) colours. As centroid colours are actual petal colours with measured colorimetric parameters, the colours of rose petals proved to be classifiable by colorimetric calculations. Confirming the expectations, it was possible to formulate a set of colorimetric conditions that ensures that the centroid colours more or less evenly fill the part of the colour space in which the rose petal colours can occur, and at the same time optimises the number of colour categories (rules 4–6). As the database of the Budatétény Rose Garden could be filled with the petal colour names determined by calculations, even rules 6 and 7 seem realisable (regulated nomenclature and algorithm-based automated classification). See the schematic diagram of the development of the colour system in Figure 1.

The viability of the petal colour system is proven by the fact that all the petal colours of the 1060 measurable cultivars and taxa of the Budatétény Rose Garden that were measured in four years could be classified and not a single petal colour was found to be unclassifiable. However, the petal colour of the varieties seems to be slightly different depending on the year and the current temperature, and sometimes the same variety had to be classified into a different category every year. However, controlled conditions certainly improve the objectivity of the classification.

The authors hope that the colorimetric criteria system developed for rose petals will be adaptable for other ornamental plants, and that the results will be suitable as a template model for new petal, foliage, or fruit colour systems for different plants, not only for roses.

## 4. Materials and Methods

### 4.1. Location

All the data were recorded in Budapest, in the Budatétény Rose Garden (Park utca 2., H-1223 Budapest, Hungary). However, in 2004, preliminary tests were also carried out in the breeding garden of the Hungarian hybridiser Gergely Márk, in Törökbálint (Malom dűlő 1., H-2045 Törökbálint, Hungary) until the garden was closed.

### 4.2. Time

The instrumental petal colour data recordings were carried out in 2018 from May 7 to 16 and from July 3 to September 15, and in 2019 from April 30. In 2020, the season of data recording was from May 18 to July 19. The dataset of 2018 mirrors the petal colours of the roses in the middle of summer, while the ones of 2019 and 2020 were recorded in the first, main blooming wave of roses (May–first half of June). Each year, the earliest data show the colouration of the once-blooming items (wild taxa and old garden roses), while the latter ones record the petal colour of the remontant roses. In addition, 67 special petal colours were recorded in 2021 between June 24 and July 9, because these rose specialities were too young in 2020.

### 4.3. Weather

The colour data were recorded on clear sunny days, when the maximum daily air temperature was between 22 and 32 °C, in order to avoid the anthocyanin concentration that usually occurs in cool weather and the anthocyanin degradation (fading) that can be observed during heat waves. For this reason, several measured colour data from the second half of September had to be excluded.

### 4.4. Instrument

All the colorimetric measurements were performed with a Konica-Minolta 600d spectrocolorimeter, where the standard of the illumination was D65, the observer standard was 10°, the geometry of the optical system was 8° diffused illumination, and the specular component was excluded (SCE). According to these standards, the recorded colorimetric parameters of the petals mirror the colouration in full sunlight, and the gloss of the petals during the measuring was set to minimal. The rose petals were laid on a thin white plastic sheet to eliminate the translucency of the petals. The petal colours were measured while the petals were lying on the card. The used material was Signe Nature Pikpot Laser 10. The colour of the card was 96.6/0.25/−3.57 in the CIE L*a*b* system, which faded to 94.4/0.34/−1.19 from the petal wax until replacement (approximately every 1500 measurements). The colour difference between the new and the worn-out card was 1.32 ΔE_00_ in CIEDE2000 standard.

### 4.5. Sampling

All the colour data used were based on instrumental measurements; visual estimation was only allowed when the petal colours were compared with RHS colour charts. This process helps to visualise the colours but has no role in the calculations (Table A3).

The petal colours could only be measured if the size of the rose petal exceeded 10 mm, because the diameter of the measuring head of the Konica-Minolta 600d spectrocolorimeter is 8 mm. According to this limitation, the number of the measured taxa of the rose collection was 1060. The number of the measured colours was 8139, which is the average of 80.330 individual measurements (after data exclusion).

The colour of the just-opened flower was measured for three years, which means three repetitions and 30 measured samples, while the additional measurements (bud colour, very young flower) were measured in one year (10 samples), considering that the goal was not to evaluate the difference between the varieties, but to map the possible petal colours.

The colours of the adaxial and abaxial surfaces (upper/inner side and under/outer side) of the petals immediately after flower opening were recorded every year, as this colouration is the most characteristic and most important of the flowers. According to Boronkay and Jámbor-Benczúr [41], this is the phenological stage 6 of the rose flower (Figure 5). This stage is characterised by the fact that both the stamens and the pistils are already differentiated and functional, and none of the stamens have dried yet. If the variety was multi-coloured, 2 colours on the abaxial surface and 3 colours on the adaxial one were studied.

Since some transition between colours on a multi-coloured petal is always noticeable, more than two or three distinguishable colours can be seen on such a petal. Therefore, the colours of the end-points of the colour transition were measured, and on the adaxial surface, the middle of the colour transition was recorded also. The basal spot was excluded, as the 42–44 paragraphs of the UPOV Guidelines for the Conduct of Tests for DUS TG/11/8 recommended [5]. In the case of monochrome petals, only one colour on each side was measured from the centre of the petal.

Each measured colour is an average of 10 individual samplings unless some data had been excluded. Since it matters in which colour system the averaging is performed, CIE L*C*h* was chosen instead of CIE L*a*b* because the averaged colour is usually more saturated if the previous polar coordinate system is used. For easier averaging, a modified hue parameter was developed called h_33_* (see Section 4.9).

In 2018, in addition to stage 6, the abaxial petal surface of the matured but still closed bud was measured also, which is the phenological substage between 3 and 4 (as value: 3.5) of the rose flower [41]. To determine the most vivid rose colouration possible (the so-called potential colour), in 2019 the adaxial surface of the petals just before opening (phenological stage 4 [41], Figure 5) was measured also. The significance of this data is given by the fact that stage 4 is the last phenophase of the rose flower when the petals virtually do not show any fading at all. The colour space formatted by these petal colours is wider than the colour variability of the open flowers of the currently existing rose varieties. These colours can be considered potential colours, because there is a chance that such a highly saturated cultivar will appear, or it already exists, but is unknown by the authors.

In addition, in some special cases, the fading colour of the petals was measured also at the very beginning of the wilting stage (phenological stage 7 [41], Figure 5), but only if it was really characteristic of a cultivar. Such are the so-called blue roses (“Rosie-Marie Viaud”, “Nuits de Young”, “Cardinal de Richelieu”) where the bluish colour can be seen only on the wilting petals, or the slowly developing purplish-brown colouration of russet roses (“Hot Cocoa”, “Edith Holden”, or “Cinco de Mayo”).

### 4.6. Data Exclusion

All the measured data were considered incorrect if the colour spectrum of the raw data gave zero value at any wavelength (Konica-Minolta 600d has 40 measuring wavelengths), because it was the result of incorrect instrument use. After this screening, the measured colours (10 samples/colour) were checked for normal distribution. Based on practical experience, the colour was considered “probably incorrect” when σ > 4 (standard deviation) and |Sk| > 2.5 (standard skewness) were found in the case of the CIE L* and C* colorimetric parameters. In the case of the hue (CIE h*) σ > 8 and |Sk| > 5 were the limitation values, as the hue parameter seemed more uncertain. As much data were excluded as needed to recover the balance of the distribution.

If the reason for the high standard deviation or skewness was the colour transition, the data of these colours were accepted because this phenomenon is very frequent on petals. If unusual colorimetric data were found which could not be rose petal colour (the measured point was possibly spot disease, fibro-vascular bundle or hair), it was excluded as an outlier. When the number of non-excluded data within a measured colour dropped below 5, the entire data set of colours was excluded. Based on the rules above, 1.85% (2018), 1.49% (2019), and 0.69% (2020) of the measured data were excluded.

### 4.7. Criteria System for Choosing a Colour for a New Centroid

In case a new category had to be added to the system and several petal colours seemed suitable, a set of criteria was developed to select the optimal one. These are the following (in descending order of importance): that colour is more suitable (A) where the colour differences from the adjacent centroid colours are closer to the 6 ΔE_00_ value; (B) where the variety is better known: it is commercially available or once it was popular; (C) where the standard deviation of the parameters of the colour is the smallest; (D) where the colour was measured on the adaxial surface of the petal. If any of the mentioned characteristics were extremely high or low, they would be considered with greater weight (e.g., a variety is very well known). The centroid colours are ideally recorded on such petals, which are monochrome, and the flower has just opened (phenological stage 6 [41], Figure 5).

In some cases, the optimal centroid colour could not be measured under the above conditions, only in a so-called atypical location or phenology stage. In this case, not only the cultivar name of the reference was indicated but the measurement condition also, e.g., “measured on the collar of young flower”. In Table 3, these special conditions can be seen.

### 4.8. Colour Standards, Colour Spaces and Colour Difference Standards

The colorimetric data provided by the instrument were in the CIE (Commission Internationale de l’Eclairage) standard: CIE L*a*b* (or CIE Lab) colour model. Here L*: lightness, a*: green-red, and b*: blue–yellow axis. The data were converted into the polar coordinate version of this standard: CIE L*C*h* (or CIE LCh) because this is perhaps the easiest-to-interpret colour standard. The parameters of the system are very close to human colour perception, as here L*: lightness; C*: colour saturation; h*: hue (in degrees). The function of this conversion is standardised: L* = L*; C* = (a^2^ + b^2^)^1/2^; h* = arc tan (b*/a*) [42].

Since CIE L*a*b* (like most colour models) describes colours with three independent dimensions—without redundancy—a colour can be represented as a point in a 3D Cartesian coordinate system. Accordingly, the set of visible colours in such a model covers a continuous part of space, which is mostly spherical. Due to the fact that the visible colours can be well represented in a 3-dimensional space, the term “colour space” will be used in the following, as it is more expressive than “colour model” or “colour standard”.

The colour difference was measured by the current, very complex, non-linear CIEDE2000 (or CIE ΔE_2000_) standard [43]. In this standard, the dimension of the colour difference is ΔE_00_. The linear 1976 CIE colour difference standard was also used for modelling, where the dimension of colour difference is ΔE_76_ using Pythagoras’ theorem: ΔE_76_ = ((L*_2_ − L*_1_)^2^ + (a*_2_ − a*_1_)^2^ + (b*_2_ − b*_1_)^2^)^1/2^.

### 4.9. Modified CIE Hue Parameter

For an easier averaging of the colour parameter measured in degrees and for a better simulation of colour perception, a modified hue parameter was created and calculated under the name h_33_* [44]. It was used for averaging the measured samples, clustering the colours into groups and representing the reference colours. The algorithm for this conversion is as follows: h’ = h* − 33°; and if h’ > 180° then h_33_* = h’ − 360° otherwise h_33_* = h’. As a result (Figure 6), the modified hue value of the “clear” red colour (considered to be neither warm nor cold) is near h_33_* = 0°, and the cold colours (blue, violet, and purple) have negative values (e.g., −1° instead of 359°), while the warm colours (orange, yellow, and green) have positive value, so the difficult-to-interpret 360° = 0° break is eliminated. However, it creates a −180° = +180° break, but this logical leap is found only in turquoise colours, which is very rare among the plants. In all other respects, the official CIE h* and h_33_* are identical. The 33° rotation, however, is not obvious and needs some more explanation.

As the range of the hue of rose petals spreads from bluish-purple to greenish-yellow, the neutral red seems to be the best for a 0° central value. However, the definition of clear red is different from standard to standard; for example, in Natural Colour System (NCS) it is L* = 41.25; a* = 66.82; b* = 30.69, and in the Munsell is L* = 50.92; a* = 78.34; b* = 38.87 in CIE L*a*b* (D65 and 10° observer). To solve this contradiction, printed colour standards were used for choosing the most clear or neutral red. Visually the “19—Scarlet” and “719—Signal Red” colours of the British Colour Council colour standard [7] were considered neither warm nor cold. In the case of “19—Scarlet”, h* = 33.8° was measured while the hue of “719—Signal red” was 32.9° h*, both in CIE L*C*h* standard. Accordingly, PANTONE Formula Guide Solid Coated “1788 CVC” (h* = 32.4°) [11] and RHS colour chart “44A” (h* = 32.9°) [6] were also good samples for the “clear” red. According to the data, the hue value of the neutral red in CIE L*C*h* is located somewhere around h* = 33°, so −33° rotation seemed to be the best for calibration.

### 4.10. Software

For the extremely complex CIEDE2000 calculation, the “Colour Conversion Centre V4.0c” online available software (Excel 2000 sheets) [45] was used. It was created by one of the authors, and its CIEDE2000 calculation is based on the work of Sharma et al. [46]. The authenticity of this software is indicated by numerous scientific works where it was used. For example, Paulson [47] used it for computing the colouration of kingfisher feathers, and Day [48] used it in the study of the preservation of turtle shells. Also, this software was cited by Taylor et al. [49], Nastiti et al. [50], El Halim et al. [51], and Ureña et al. [52]. They studied (in the order of the list) perfluorocarboxylic acids, chicken breast meats, igneous and sedimentary rocks, and olive oils.

Sometimes very large CIEDE2000 matrices had to be generated to clarify the relationships between petal colours, for this reason, an automatic matrix generation software was also developed, called “CCCAutoMatrix 1.0”, which works with the algorithms of the Colour Conversion Centre, and it is also available online [53].

The exclusive function of the algorithms of the CIEDE2000 worksheet of the Colour Conversion Centre (based on Sharma et al. [46]) is to calculate CIEDE2000 type colour difference between two colours defined in the CIE L*a*b* colour standard. The inputs of the Colour Conversion Centre and CCCAutoMatrix are the CIE L*a*b* parameters of the colours between which the colour distance must be measured, and the weight factors for the L*, a*, and b* parameters. However, these factors should be set as 1:1:1 (default values). The output is the colour distance in ΔE_00_ as a value.

To create colour groups, the hierarchical cluster analysis of Statgraphics Centurion V.18 was used with Ward’s minimal variance method, as this seemed the most appropriate for forming groups similar in size. For clustering the centroid colours, five CIE parameters were used as data variables: L*, a*, b*, C*, and h_33_* (modified h*, see Section 4.9). The parameters of the hierarchical cluster analysis were the following: the difference metric was squared Euclidean, and the variables were standardised.

### 4.11. Printed, Paper-Based Colour Standards

In order to describe the colours of the categories in a way that is easy to understand and reproduce, the centroid colours were identified with printed colour cards also. For this purpose, the latest edition of the colour standard of UPOV was used, which is the “RHS Colour Chart Sixth Edition” [6]. To describe a petal colour, a visual comparison was used under controlled conditions. The comparison was always made on sunny days except for midday, and both the colour cards and the petals were held 90° from the sun (usually it meant a western direction) to exclude the velvety effect of the petals and the glitter effect of the paper.

The colours of the colour chart are indicated by number–letter combinations. When a card colour was identical to a petal colour, it was described by the number–letter combination of the card (e.g., #49 Begonia coral is RHS 40c). If the average of two RHS colours gives an acceptable perception of the petal colour, in the description both card codes are included, separated by “/” sign (e.g., #57 Capsicum red is RHS N30a/N30b). Often this approximation was insufficient, and extrapolation was needed. In such cases, the estimated difference (addition or subtraction) was noted in the CIE L*C*h* system. For example, the code of #58 vermilion red in this system is N30a/40a C* + 10 because this colour is somewhere between the RHS colours N30a and 40a; however, the colour is more saturated by about 10 chroma units in CIE L*C*h* standard.

### 4.12. Colour Names

The following rules were used when a category was named: Each colour should consist of a basic colour name in spoken language and an adjective. The adjective should not be the “dark/deep/bright/light, etc.” kind of indefinite expression. The adjective must be unique, so any adjective should be used only once. According to this rule, in this system, Spanish orange and Spanish pink or Coral pink and Coral red are not allowed at the same time. All colours should already be used by at least one industry colour standard. Where it was possible, the colour names refer to the chemical substance from which the paint is made or natural minerals, plants, and animals because they are known by very different cultures, and they are reliable as references (Table A1 and Table A3). However, this strict system was not applied to the names of colour groups, because grouping is not an integral part of this colour system, and the number of groups can be changed freely.

The naming process was based on the following principles: that name is more suitable: (1) when the colour name describes the centroid colour more correctly; (2) when the colour name is better known; (3) when the colour name is more objective, i.e., fancy names must be omitted; (4) when the colour standard where the name was found is more accepted.

Only a few industrial colour standards were found where both colour names and colorimetric parameters could be found. Moreover, the exact CIE L*a*b* parameters of industrial colour collections are usually available only on unofficial web pages, which means that the reliability of this data is limited. The evaluated colour systems (Table A2) are the following: HTML (HyperText Markup Language) colours [36], the Crayola standard colours [37] of Crayola LLC (Easton, PA, USA), the Pantone TCX colours [38] of Pantone^®^ LLC, and the RAL Classic colours and RAL Design System+ [39] of RAL GmbH.

### 4.13. Names of Cultivars and Wild Taxa

For easier interpretation in this publication, the names of the rose cultivars are the American Exhibition Names (AEN) of the American Rose Society [3] which are trade names, not code names (designated variety denominations). The source of the Latin names is the online Catalogue of Life—Annual Checklist [54].

## Figures and Tables

**Figure 1 plants-13-01368-f001:**
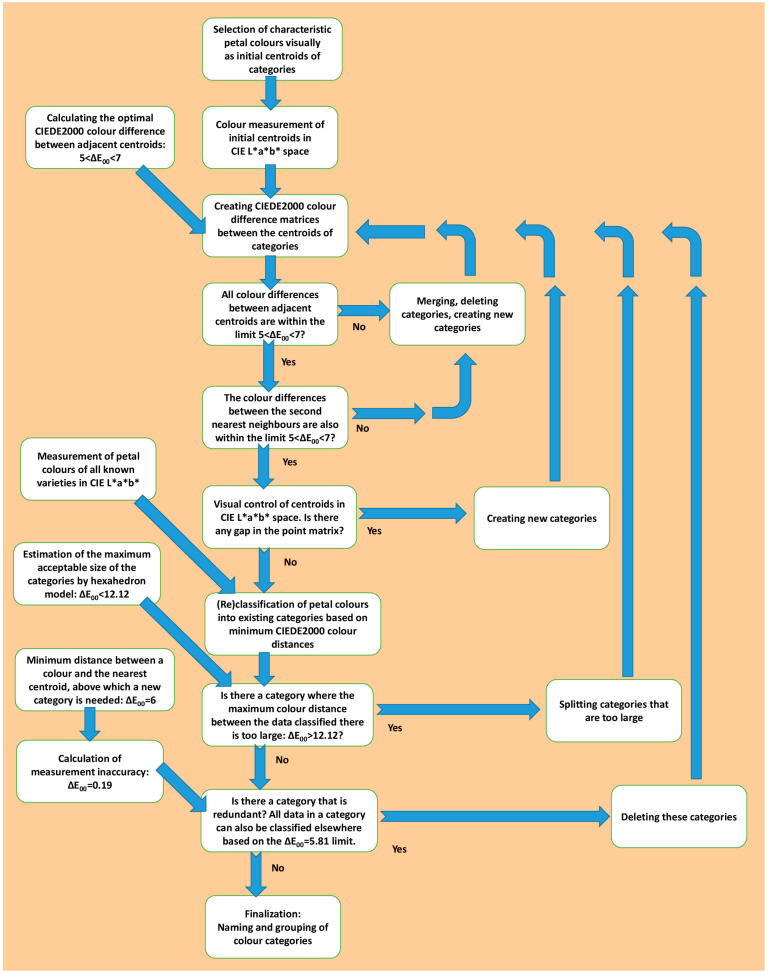
Schematic diagram of the development of the rose petal colour system.

**Figure 2 plants-13-01368-f002:**
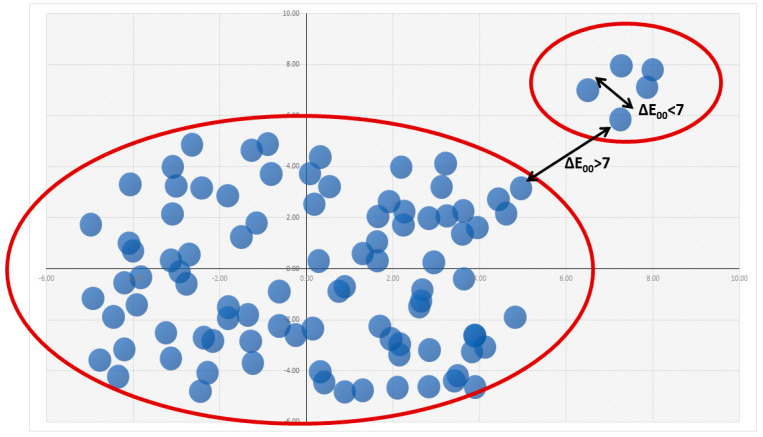
Illustrative example: Even if each colour category has at least one neighbour within the 5 < ΔE_00_ < 7 colour difference, an incomplete network could develop with gaps between the colour subgroups. Here, each point refers to a centroid colour in a colour space (simplified to 2 dimensions).

**Figure 3 plants-13-01368-f003:**
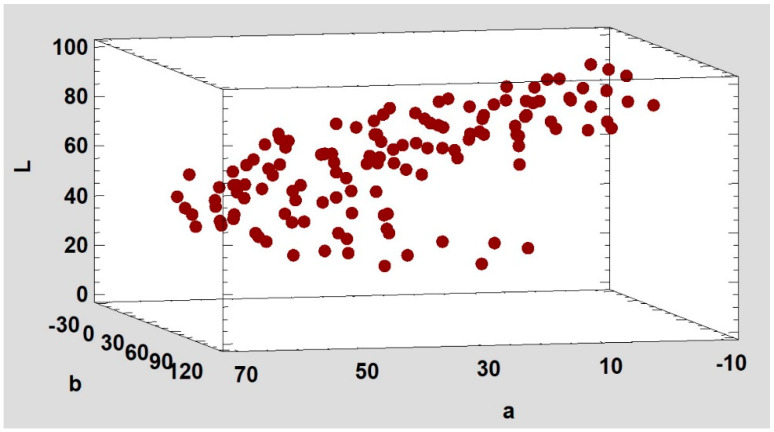
The 3D arrangement of 133 centroid colours of the petal classification system in the CIE L*a*b* colour space. The image is a snapshot from a video where the set of points rotates around the axes to search for missing centroids.

**Figure 4 plants-13-01368-f004:**
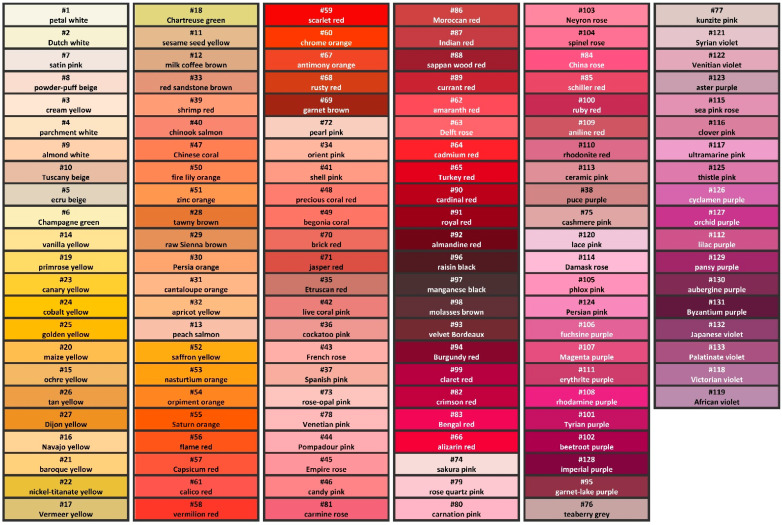
The colours of the rose petal classification system in colour harmony order. The numbering system and the recommended names by the authors are shown. For printability, the CIE L*a*b* parameters of the colours were converted to sRGB, and lightness and saturation were raised by 10%.

**Figure 5 plants-13-01368-f005:**
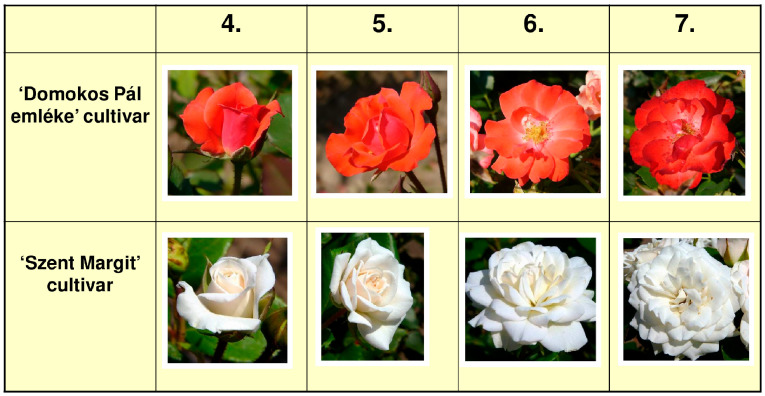
The numbering system of the phenophases of the young rose flowers according to Boronkay and Jámbor-Benczúr [41]. These cultivars are Hungarian roses bred by G. Márk.

**Figure 6 plants-13-01368-f006:**
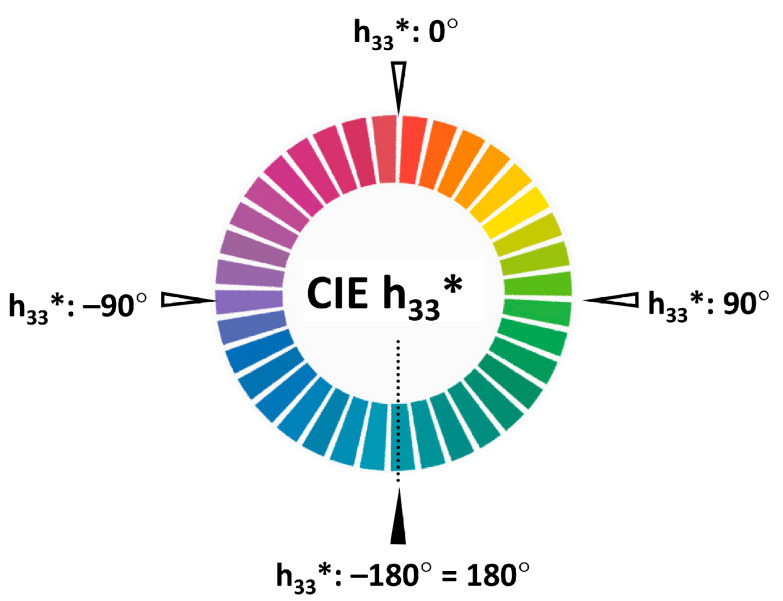
Visual implementation of the hue dimension of the CIE L*C*h* colour space modified by the authors, where h_33_* replaces the original h* hue parameter.

**Table 1 plants-13-01368-t001:** CIEDE2000 colour differences between some visually well-recognisable rose petal colours, selected in order to calculate the optimal colour difference and the RHS colour chart codes of these colours Marking of RHS colours is according to Section 4.11 “Materials and Methods”.

	Yellow	Pink	Red
Very light	RHS: 8D	RHS: 36D	RHS: 44D
Difference between very light and light colours	7.2 ΔE_00_	6.4 ΔE_00_	3.1 ΔE_00_
Light	RHS: 8C	RHS: 36B	RHS: 41B
Difference between light and medium colours	5.8 ΔE_00_	6.1 ΔE_00_	3.2 ΔE_00_
Medium	RHS: 9C	RHS: 49B/49C	RHS: 40A/40B
Difference between medium and dark colours	5.8 ΔE_00_	8.6 ΔE_00_	8.4 ΔE_00_
Dark	RHS: 12A	RHS: 38A/38B	RHS: 44B

**Table 2 plants-13-01368-t002:** Distribution of colour categories recorded in 2018–2021, in Budatétény Rose Garden (Budapest, Hungary) in descending order of percentage.

Frequency	Suggested Names of Colour Categories
>3.00%	#83 Bengal red
2.75–3.00%	#1 petal white; #103 Neyron rose; #85 schiller red; #90 cardinal red
2.50–2.74%	-
2.25–2.49%	#80 carnation pink; #2 Dutch white
2.00–2.24%	#107 Magenta purple; #89 currant red; #79 rose quartz pink
1.75–1.99%	#100 ruby red; #21 Baroque yellow; #82 crimson red; #14 vanilla yellow; #66 alizarin red; #84 China rose; #91 royal red
1.50–1.74%	#63 Delft rose; #62 amaranth red; #106 fuchsine purple; #23 Canary yellow; #105 phlox pink; #104 Spinel rose; #74 Sakura pink; #81 carmine rose
1.25–1.49%	#19 primrose yellow; #4 parchment white; #115 sea pink rose; #78 Venetian pink; #73 rose opal pink; #64 cadmium red; #32 apricot yellow; #61 calico red; #46 candy pink
1.00–1.24%	#20 maize yellow; #44 Pompadour pink; #65 Turkey red; #3 cream yellow; #92 almond white; #8 Powder puff beige
0.75–0.99%	#16 Navajo yellow; #49 begonia coral; #58 vermilion red; #72 pearl pink; #7 satin pink; #45 Empire rose; #88 sappan wood red
0.50–0.74%	#114 Damask rose pink; #57 Capsicum red; #6 Champagne green; #92 almandine red; #31 Cantaloupe orange; #25 golden yellow; #13 peach salmon; #124 Persian pink; #43 French rose; #39 shrimp red; #59 scarlet red; #34 orient pink; #121 Syrian violet; #116 clover pink; #111 erythrite purple
0.25–0.49%	#99 claret red; #41 shell pink;#51 zinc orange; #122 Venetic violet; #52 saffron yellow; #47 Chinese coral; #108 rhodamine purple; #112 lilac purple; #37 Spanish pink; #94 Burgundy red; #24 cobalt yellow; #113 ceramic pink; #53 nasturtium orange; #36 cockatoo pink; #101 Tyrian purple; #110 rhodonite red; #48 precious coral red; #40 chinook salmon
0.10–0.24%	#15 ochre yellow; #50 fire lily orange; #77 kunzite pink; #35 Etruscan red; #26 tan yellow; #56 flame red; #117 ultramarine pink; # velvet Bordeaux; #86 Moroccan red; #55 Saturn orange; #96 raisin black; #30 Persia orange; #120 lace pink; #60 chrome orange; #54 orpiment orange; #132 Japanese violet; #71 jasper red; #131 Byzantium purple; #128 Imperial purple; #129 pansy purple; #109 aniline red; #127 orchid purple; #130 aubergine purple; #42 live coral pink; #125 thistle pink; #123 aster purple
<0.10%	#69 garnet brown; #133 Palatinate violet; #87 Indian red, #102 beetroot purple; #5 ecru beige; #12 milk coffee brown; #33 red sandstone brown, #29 raw Sienna brown; #75 cashmere pink; #11 sesame seed yellow; #119 African violet, #10 Tuscany beige; #95 garnet lake purple, #126 cyclamen purple, #18 Chartreuse green, #27 Dijon yellow; #67 antimony orange; #68 rusty red; #118 Victorian violet; #28 tawny brown; #70 brick red, #38 puce purple; #76 teaberry grey, #22 nickel-titanate yellow; #17 Vermeer yellow; #98 molasses brown; #97 manganese black

**Table 3 plants-13-01368-t003:** Typical and atypical locations of centroid colours: places on the petal and the phenophases of the flower. If a centroid colour was measured under atypical conditions, it should be indicated.

	Place of Measurement on the Petals	Measured Phenophases of the Flower [41]
Typical (unmarked)	the middle of the adaxial (upperside) surface of the petal	just opened flower (phenophase 6)
Atypical	the abaxial (underside) surface	developed bud (phenophase 3.5)
	the adaxial surface of the collar (outermost petal)	bud just before opening (phenophase 4)
	the abaxial surface of the petal of the closed bud	young flower (phenophase 5.5)
	the adaxial surface of the petal of the opening bud	fading petal, at the beginning of wilting (phenophase 7)
	the edge of the adaxial surface of the petal	
	the edge of the adaxial surface of the outermost petal	

## Data Availability

The data presented in this study are available on request from the corresponding author.

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
