# Peer review of "Developing a Colorimetrically Balanced, Measurement-Based Petal Colour System for Cultivated Rose (Rosa L. Cultivars) and the Resulting Colour Categories"

_plants, 2024, doi:10.3390/plants13101368_

Round 1

Reviewer 1 Report (New Reviewer)

Comments and Suggestions for Authors

The submitted paper proposes a  colorimetric measurements and categorization protocol for petal colors.

The introduction is well written, as the problem/aim of the study is appropriately stated and supported by appropriate bibliography. The Materials and Methods section fully describes the work that was done, experimentally and in terms of data manipulation also. The results are shown and discussed in details.

My first thought regarding the proposed method was somewhat negative, as I believe that in-situ measurements with portable spectrophotometers are not objective; I prefer that this kind of measurements should be performed with laboratory spectrophotometers that are equipped with integrating spheres. In this case, the samples are measured in stable and dark conditions, and the environment's light doesn't affect the measurements. Nevertheless, the authors seem to take this under consideration, and they concluded to a appropriate measurement protocol. It should also be mentioned, that the authors performed a huge amount of measurements, for a prolonged time period.

From my point of view, the manuscript under review should be considered for publication with Plants.

I propose that the authors should change their abstract a little, following a much more
descriptive structure (state of the problem → method and materials → most significant results).

Comments on the Quality of English Language

1.       Please, check the Journal’s language requirements regarding the use of UK or USA English, in order to use “color” or “colour”

2.       Line 82: “focused”

3. Please, use plural after "data"

4. Although the manuscript is well-written in terms of language, please, check the whole document for minor errors, like singular-plural or need from active to passive voice changes.

Author Response

Manuscript ID: plants-2936515
Title: Developing a colorimetrically balanced, measurement-based petal colour system for cultivated rose (Rosa L. cultivars) and the resulting colour categories
Authors: Gábor Boronkay *, Dóra Hamar-Farkas, Szilvia Kisvarga, Zsuzsanna Bekefi, András Neményi, László Orlóci

From:

Dr. Gábor BORONKAY

MATE Institute of Landscape Architecture, Urban Planning and Garden Art (MATE ILAUG)

Hungarian University of Agriculture and Life Sciences (MATE)

Ornamental Plants and Green System Management Research Group

  1. Park utca, Budapest, Hungary, H-1223

boronkay.gabor@uni-mate.hu

Response for reviewer I.

Thank you for your review and also for your very thorough check. We appreciate your work that greatly contributes to our work.

Allow me to inform you about the changes recommended by you.

I propose that the authors should change their abstract a little, following a much more descriptive structure.

Here is the copy of the corrected Abstract:

There is no practical and at the same time objective colour system available for describing cultivated roses (Rosa L. cultivars). For this reason a new colour classification system was developed which is colorimetrically balanced, appropriate for algorithmic colour identification, however it is also suitable for field-work. The system is based on the following colorimetric criteria: A) Each colour category is characterized by a measured petal colour in the CIE L*a*b* standard as the centroid of the category. B) The CIEDE2000 colour differences between the adjacent centroid colours are limited (5 < ΔE00 < 7). C) The maximal colour difference between the measured colours in a category is also limited (to 12.12 ΔE00). D) A measured petal colour can only be classified into an existing category if the colour difference from the centroid colour of the given category is less than 5.81 ΔE00, otherwise a new category is required. E) A category is only considered as non-redundant, if it has at least one measured petal colour that cannot be classified elsewhere. F) The classification of the petal colours is based on the least colour difference from the centroid colours. As a result, 133 colour categories were required for describing all the 8139 petal colours of the rose cultivars of Budatétény Rose Garden (Hungary). Each colour category has the following parameters: standardized colour name, colorimetric parameters of the centroid, grouping, RHS colour chart coding and reference cultivars, which are described in the article.

Comments on the Quality of English Language:

We tried to prepare the entire text according to the UK dialect and spelling. We checked the text, and corrected the grammatical errors which the reviewer noticed. Also the text was checked by a college who is native speaker of British English, and we took his recommendations into account as well.

Dr. Gábor Boronkay

Budapest, 2024. April 23.

Reviewer 2 Report (New Reviewer)

Comments and Suggestions for Authors

In the fields of botany and horticulture, the classification and naming of flower colors have always been topics of great interest. Traditionally, this task relied primarily on subjective observation and personal experience, lacking objective standards and rules. However, with the development of colorimetric calculation technology, people began to explore the use of objective numerical and computational methods for classifying and naming flower colors. This article introduces the development process and implementation of a color classification system for rose petals based on colorimetric calculations. This system provides a new method for flower color classification and has achieved satisfactory results in practice.

The distinguishing feature of this rose petal color classification system lies in its innovation and practicality. By combining colorimetric calculations with visual classification, the system can objectively and accurately classify the colors of rose petals, avoiding the subjective biases and inconsistencies inherent in traditional classification methods. Furthermore, the system employs mathematical methods such as cluster analysis to group color categories into larger color groups, making the system more comprehensive and easy to understand. Therefore, this system is not only innovative in theory but also highly practical in practice, providing strong support for floriculture classification and horticultural practices.

The rose petal color classification system described in this article demonstrates excellent reliability and accuracy. By visually classifying and colorimetrically calculating a large number of rose varieties, the system can accurately identify and classify the color of each flower. Moreover, through cluster analysis and other mathematical methods, the system can to some extent correct errors caused by data variability, thereby improving the reliability of the classification results. Therefore, this system not only has high accuracy but also can be reliably applied in different datasets and practical scenarios.

It is worth noting that although the rose petal color classification system described in this article was primarily developed and tested for the color of rose petals, its practical application may be far broader. Due to the universality and scalability of colorimetric calculation technology, this classification system can theoretically be applied to the color classification of flower petals, leaves, and even fruit of other plants. Therefore, this system provides a new perspective and method for botanical taxonomy and horticulture research, with broad application prospects.

In practical applications, the rose petal color classification system has been validated in horticultural practice and proven to be an effective tool. However, as the system is further promoted and applied, there are still many potential directions for development and improvement. For example, algorithms can be further optimized to increase the system's automation and efficiency; the system can be applied to the color classification of other plants, establishing a universal plant color classification system; and machine learning and artificial intelligence technologies can be combined to further improve the system's accuracy and intelligence. Therefore, although the rose petal color classification system has achieved significant results, there are still many challenges and opportunities to explore and realize in the future.

In summary, the evaluation of the rose petal color classification system based on colorimetric calculations demonstrates its importance and value in the fields of botany and horticulture. Firstly, the innovative and practical nature of this classification system represents a significant breakthrough in floriculture classification. Traditional flower color classification relies on subjective observation and experience, which poses subjective biases and inconsistencies. However, the colorimetric-based system offers an objective and accurate alternative, contributing significantly to the development and progress of botanical color classification. Secondly, the reliability and accuracy of the system have been thoroughly validated, ensuring its applicability across various datasets and practical scenarios. Lastly, the scalability and applicability of the system open up new avenues for research and application in botanical taxonomy and horticulture, promising a brighter future for plant color classification studies. Therefore, the system not only holds theoretical significance but also possesses high practical value, making substantial contributions to the advancement of botanical color classification.

Author Response

Manuscript ID: plants-2936515
Title: Developing a colorimetrically balanced, measurement-based petal colour system for cultivated rose (Rosa L. cultivars) and the resulting colour categories
Authors: Gábor Boronkay *, Dóra Hamar-Farkas, Szilvia Kisvarga, Zsuzsanna Békefi, András Neményi, László Orlóci

From:

Dr. Gábor BORONKAY

MATE Institute of Landscape Architecture, Urban Planning and Garden Art (MATE ILAUG)

Hungarian University of Agriculture and Life Sciences (MATE)

Ornamental Plants and Green System Management Research Group

  1. Park utca, Budapest, Hungary, H-1223

boronkay.gabor@uni-mate.hu

Response for reviewer II.

Thank you for your review and also for your very thorough check. We appreciate your work that greatly contributes to our work. I would also like to thank you for the words of praise and the fact that you consider our work valuable in a wider context.

Allow me respond to comments:

... Therefore, although the rose petal colour classification system has achieved significant results, there are still many challenges and opportunities to explore and realize in the future....

Indeed, this system can be improved, in some ways. Recently the authors are working on a new colour classification system based on this one, which will be able to describe the colour of the foliage of the rose cultivars. In this system the initial step will be no more a visual selection of the most characteristic colours, but it will be based on a Principal Component Analysis, as for leaves it is difficult to define any characteristic colour. However this change does not affect the results, only helps to start the work.

According to our long-term plans, we will apply our idea to bamboo and possibly other ornamental shrubs as well, if the horticultural community accepts such a colour system.

Dr. Gábor Boronkay

Budapest, 2024. April 23.

Reviewer 3 Report (New Reviewer)

Comments and Suggestions for Authors

The article seems to have interesting data, and there is no doubt that it involves immense work. However, I have doubts about the need for so many industrial colour classes if the human eye cannot differentiate between so many classes.

On the other hand, the text is confusing, and does not allow for fluent and understandable reading. Maybe the inclusion of some diagrams to better understand how all the steps were carried out, can bring benefits

It is not clear why such a wide range of colours is necessary.

The authors mention that "only the most characteristic colours of the petals were measured". The criteria for deciding this step are not specified in the manuscript, but generally, when determining the criteria for measuring the most characteristic colours of the petals, it is important  consider factors such as the frequency of occurrence of the colours, the representativeness of the colours in relation to the diversity of the species or variety of rose studied 8analysed in this paper but do not present some results to better understand), the aesthetic or commercial importance of the colours, among others. These criteria may vary depending on the purpose of the study and I think that must be well explained or defined. The aim of the study is not clear identified.

The authors mention algorithms for colour analysis, but there is a lack of explanation regarding which algorithms were used and how they input the data and differentiate the outputs.

Author Response

Manuscript ID: plants-2936515
Title: Developing a colorimetrically balanced, measurement-based petal colour system for cultivated rose (Rosa L. cultivars) and the resulting colour categories
Authors: Gábor Boronkay *, Dóra Hamar-Farkas, Szilvia Kisvarga, Zsuzsanna Békefi, András Neményi, László Orlóci

From:

Dr. Gábor BORONKAY

MATE Institute of Landscape Architecture, Urban Planning and Garden Art (MATE ILAUG)

Hungarian University of Agriculture and Life Sciences (MATE)

Ornamental Plants and Green System Management Research Group

  1. Park utca, Budapest, Hungary, H-1223

boronkay.gabor@uni-mate.hu

Response for reviewer III.

Thank you for your review and also for your very thorough check. We appreciate your work that greatly contributes to our work.

Allow me to respond to the errors you have noticed. Accepting the critical comments, I would like to present the corrections below, copying the relevant parts of the revised manuscript in italics:

- However, I have doubts about the need for so many industrial colour classes if the human eye cannot differentiate between so many classes.

Comparing these 133 colour categories to industry colour standards, the number of categories does not seem too high. For instance, the Sixth Revised Edition of the RHS Color Chart (a standard specifically optimised for plant colours) provides 920 colours, although there are hard-to-distinguish colours and colour gaps can be found. However, the number of physically detectable colours actually occurring on a rose petal cannot be defined, it is practically infinite, and only the physiological processes of colour perception determine how many petal colours a person can distinguish. That is why it seemed necessary to set up objective colour categories, where the colours can be easily separated from each other, yet the system has sufficient resolution to separate the rose cultivars.

- Maybe the inclusion of some diagrams to better understand how all the steps were carried out, can bring benefits

The authors created a flowchart type diagram to make this really quite complex creative process easier to understand. Here is the copy of the chart:

- It is not clear why such a wide range of colours is necessary.

This classification needed 133 colour categories, which are presented in Tables A1, A3, A4 and Figure 4. The number of categories was determined by the colour rules used and the spectrum of the actual measured petal colours. Since Budatétény Rose Garden has a significant number of rose cultivars, several of them have special colours, and the colorimetric data were recorded even on the unopened flowers, this petal colour system covers a much wider colour range than the colorimetric variability of everyday commercial rose cultivars.

- The authors mention that "only the most characteristic colours of the petals were measured".

This sentence was corrected: Since some transition between colours on a multi-coloured petal is always noticeable, more than two or three distinguishable colours can be seen on such a petal. Therefore, the colours of the end-points of the colour transition were measured, and on the adaxial surface, the middle of the colour transition was recorded also. The basal spot was excluded, as the 42-44 paragraphs of the UPOV Guidelines for the Conduct of Tests for DUS TG/11/8 recommended [5]. In the case of monochrome petals, only one colour on each side was measured, from the centre of the petal.

- ... it is important consider factors such as the frequency of occurrence of the colours, the representativeness of the colours in relation to the diversity of the species or variety of rose studied analysed in this paper but do not present some results to better understand), the aesthetic or commercial importance of the colours, among others. These criteria may vary depending on the purpose of the study and I think that must be well explained or defined. ...

It seems surprising that there is a relatively small number of studies on the social role of rose flower colour because it seems to be a very important trait. The colour of the flower is the main consideration when buying a rose, although roses were originally cultivated for their fragrance. Shopping habits [29] were investigated in Germany, as it is the largest host market in the European Union. Here, the most common colour of solid bouquets was red and yellow (24% and 23%), but yellows and orange-salmon colours gradually became more popular compared to shades of red and pink.

The psychological value of petal colours is unique to the individual, but is strongly influenced by current fashion trends. Cultivation tries to satisfy this immediate need, so according to surveys, in addition to psychological factors, marketing and financial aspects also play a role. There was a survey in South Africa [30] that specifically looked at whether people buy different colours or types of flowers for different emotional reasons (gift, hospital visit, home). However, no clear pattern emerged from the results. Neither the type of flower nor the colour was of primary importance when purchasing. Apart from personal taste, the customers only took into account that the colour of the flower should be clean and bright. However, this was probably driven by fashion, as later soft colours (romantic) and gradient petals became popular.

- The authors mention algorithms for colour analysis, but there is a lack of explanation regarding which algorithms were used and how they input the data and differentiate the outputs.

The exclusive function of the algorithms of the CIEDE2000 worksheet of Colour Conversion Centre (based on Sharma et al. [46]) is to calculate CIEDE2000 type colour difference between two colours defined in CIE L*a*b* colour standard. The input of Colour Conversion Centre and CCCAutoMatrix are the CIE L*a*b* parameters of the colours between which the colour distance must be measured, and weight factors for the L*, a*, b* parameters. However, these factors should be set as 1:1:1 (default values). The output is the colour distance in ΔE00 as a value.

Dr. Gábor Boronkay

Budapest, 2024. April 23.

Round 2

Reviewer 3 Report (New Reviewer)

Comments and Suggestions for Authors

The authors have kindly addressed my inquiries and supplemented the article with the necessary details, the manuscript it suitable for acceptance

This manuscript is a resubmission of an earlier submission. The following is a list of the peer review reports and author responses from that submission.

Round 1

Reviewer 1 Report

Comments and Suggestions for Authors

This paper exhibits important results concerning a very long study, covering the period 2004-2022. The successive campaigns of color measurements permit to adapt and modify the general methodology. Here are presented a lot of results and a well-founded method. A large and detailed  Appendix gives a set of results and a useful classification.  The high complexity of color measurements on a set of flowers seems to be very accurate though a lot of parameters depending on time, seasons, temperature, aging, etc. are encountered.  More than 80 000 individual measurements were made ! So it was obviously necessary to find a methodology for analysis, classification, comparison. The results and different methods are well described and accessible to the non-specialist like me. An important problem is revealed about the vocabulary dedicated to flowers in general and mainly, as concerned here, for roses. The choice of the most pertinent colorimetric space, the calculations in CIEDE2000, the small transformation leading to CIE h33, make the results more comprehensive and easy to compare themselves. I nevertheless found a bibliographic reference that might be of great interest for the authors :

Naming the colors: Color names designation from the colorimetric values. The French GPEM/PV work revisited

Robert Sève Color Research & ApplicationVolume 43, Issue 5

That author also gives an Excel sheet allowing to explore and define the most relevant color names being given colorimetric coordinates.

The work here submitted is considerable indeed...it cannot be ignored.

Reviewer 2 Report

Comments and Suggestions for Authors

File attached

Reviewer 3 Report

Comments and Suggestions for Authors

Authors could reconsider the structure of the manuscript.

Aim of the study should be clearly stated in Introduction with background, rather than inside of Results section. If authors may have any hypothesis, those should be described in Introduction.

Methodology should be stated before Results. More scientific and physical information would be required. Authors should have provided conclusion or summary of the study at the end.

Terminology used in the manuscript did not seem to be well defined, or different from standard colorimetric practice. Please clarify, for example, what “length of a colour…”, “chromatic distance”,  “chromatic difference limit” were defined and how “colour category” and “colour class” are related each other. What is the “psycho-chromatic space”?

Experimental conditions (measurement conditions and protocol) should be clearly given.

It was unclear how the measurements were taken place on what.

Some statements in analysis were difficult to follow. For example,

“interface” in “… the interface between two colour classes is a bounding surface where the chromatic differences to the neighbouring centroids are the same.” (subsection 2.2)

“As a result, 3D net of the colour classes was developed and each class had a centroid colour and a reference cultivar (where the centroid colour was measured). All measured colours were classified into these categories, although at this level the balance of the colour classes in the colour space was still based on personal visual perception.” (subsection 2.3)

“…and the system still contained several subjective elements from the original, visually defined classes”. (subsection 2.4)

Please clarify which part of the “measurements” were subjective judgement and were objective (physical measurements), and what criteria were used.

Figures and tables need more explanation.

Please explain how the colour naming was undertaken. What principle did you follow?

What was the basis of the colour naming?  It was unclear how authors may have followed the other studies in colour naming and colour categorisation, or whether authors wanted to develop a protocol by themselves.

Please provide a rationale to introduce a “modified CIE h* parameter”. Please explain why it was needed and what is the basis of the 33 deg?

Authors may have had an assumption that distributions of colour cluster (or colour categories) of natural objects should be uniformly filling in a colour space.

If authors may have had any hypothesis or proposition, those should have been stated in Introduction.